# Obstructive Sleep Apnea and Nonalcoholic Fatty Liver Disease in the General Population: A Cross-Sectional Study Using Nationally Representative Data

**DOI:** 10.3390/ijerph19148398

**Published:** 2022-07-09

**Authors:** Taeyun Kim, Hyunji Choi, Jaejun Lee, Jehun Kim

**Affiliations:** 1Division of Pulmonology, Department of Internal Medicine, The Armed Forces Goyang Hospital, Goyang 10271, Korea; jimsb89@naver.com; 2Department of Laboratory Medicine, Industry-Academic Cooperation Foundation of Kosin University, Busan 49104, Korea; woojukirin@naver.com; 3Division of Hepatology, Department of Internal Medicine, The Armed Forces Goyang Hospital, Goyang 10271, Korea; pwln0516@gmail.com; 4Division of Pulmonology, Department of Internal Medicine, Kosin University Gospel Hospital, Busan 49267, Korea

**Keywords:** obstructive sleep apnea, STOP-Bang, NAFLD, fibrosis, KNHANES

## Abstract

(1) Background: To evaluate the association between obstructive sleep apnea (OSA) and nonalcoholic fatty liver disease (NAFLD) in the general population using a nationally representative sample from South Korea; (2) Methods: This study included 4275 adults aged ≥40 years who completed the snoring, tiredness, observed apnea, high blood pressure, body mass index (BMI), age, neck circumference, and gender (STOP-Bang) questionnaire. The risk of OSA was stratified into low, intermediate, and high grades according to the STOP-Bang score. The prevalence of NAFLD according to the STOP-Bang score was calculated, and the increasing trend was measured. A complex sample multivariable regression analysis with adjustments for possible confounding variables was used to calculate the odds ratio of NAFLD and advanced fibrosis. Subgroup analysis was conducted with stratification based on sex and obesity status; (3) Results: We identified 1021 adults with NAFLD and 3254 adults without NAFLD. The prevalence of NAFLD increased significantly with higher STOP-Bang scores in both men and women. Participants of both sexes with high STOP-Bang scores were more likely to have NAFLD. Compared to non-obese individuals, the risk of NAFLD according to the STOP-Bang score was more intense in obese individuals. With respect to hepatic steatosis, there was no significant association between advanced fibrosis and STOP-Bang score; (4) Conclusions: OSA, the risk of which was measured using the STOP-Bang model, was closely associated with NAFLD in both Korean men and women. Clinicians should consider screening for NAFLD in individuals with a high STOP-Bang score.

## 1. Introduction

Obstructive sleep apnea (OSA) is characterized by intermittent repetitive interruptions of breathing during sleep caused by the collapse of the respiratory passage [1]. The estimated prevalence of OSA is approximately 4–14% in Asian populations and is much higher among obese individuals [2]. The prevalence of OSA increases with an increase in the obese population [3]. In addition to obesity, OSA is closely associated with several metabolic disorders, such as type 2 diabetes [4] and nonalcoholic fatty liver disease (NAFLD) [5]. OSA leads to chronic intermittent hypoxia (IH), which is a hallmark factor in the pathogenesis and exacerbation of metabolic conditions. IH causes oxidative stress, inflammatory cascades, lipotoxicity, and insulin resistance [6], all of which share a physiopathology similar to that of NAFLD. The severity of OSA is associated with advanced liver fibrosis in patients with NAFLD [7].

Various experimental and epidemiological studies report an association between OSA and NAFLD. A mouse model showed that IH contributes to increased levels of proinflammatory cytokines, suggesting a role for IH in the development of hepatic steatosis and steatohepatitis [8]. A disease code-based observational study using data from the national insurance system showed an association between OSA and NAFLD, and subgroup analyses revealed that a younger age, male sex, higher body mass index (BMI), and abdominal obesity were more related to this association [9]. However, the data claims may have underestimated the prevalence of OSA and did not measure the degree of OSA [9]. Another human study found a correlation between elevated liver enzyme levels and OSA severity and showed the efficacy of continuous positive airway pressure therapy in improving liver steatosis [10].

Despite several criticisms of the apnea–hypopnea index (AHI), it remains the most widely used tool in diagnosing OSA. Sleep studies or polysomnography (PSG) are required to estimate this index. However, because PSG is costly, technically difficult, time-consuming, and labor-intensive, several alternative approaches, such as home respiratory polygraphy have been introduced [11]. Another simple, easy-to-use method that has been validated for screening OSA is the snoring, tiredness, observed apnea, high blood pressure, BMI, age, neck circumference, and male sex (STOP-Bang) questionnaire. As a higher score correlates with a higher risk of having OSA [12], individuals can be stratified according to their STOP-Bang score. Thus, this model has been demonstrated to be a useful tool for predicting the presence of OSA across different study designs, sample sizes, validation tools, and geographical regions [13].

However, no previous study has evaluated the association between OSA and NAFLD by using the STOP-Bang scoring system in the general population. One previous study used the STOP-Bang score for assessing OSA, but only a small number of individuals were included, and it was limited to patients with NAFLD [14]. Therefore, the present study aimed to evaluate the relationship between the STOP-Bang score-stratified risk of OSA, NAFLD, and advanced fibrosis in patients using the dataset of the eighth Korean National Health and Nutrition Examination Survey (KNHANES).

## 2. Materials and Methods

### 2.1. Study Participants

The present study used the eighth (2019–2020) KNHANES dataset. A total of 20,808 individuals were selected for the survey during the study period, and 15,469 responded, with a response rate of 74.3%. Participants aged ≥40 years who completed the STOP-Bang questionnaire were selected (*n* = 8061; 3495 men and 4566 women). We excluded individuals with evidence of preceding liver disease etiologies: positive for hepatitis B surface antigen (*n* = 260), positive for hepatitis C antibody (*n* = 75), history of liver cirrhosis (*n* = 32), or heavy alcohol consumption (>210 g of alcohol per week for men and >140 g for women, *n* = 639). We also excluded individuals with missing data for the hepatic steatosis index (HSI), which was used to determine the presence of NAFLD (*n* = 1483). Finally, 4275 individuals (2001 men and 2274 women) were included in the analysis.

This study was approved by the institutional review board of the Korea Centers for Disease Control and Prevention (2018-01-03-C-A in 2019 and 2018-01-03-2C-A in 2020). Written informed consent to participate in the study was obtained from all participants.

### 2.2. The Definition of NAFLD and Evaluation of Advanced Liver Fibrosis

After excluding individuals positive for hepatitis B surface antigen, positive for hepatitis C antibody, those with a history of liver cirrhosis, and those with a history of heavy alcohol consumption, NAFLD was defined using the HSI, which is a noninvasive, validated method for determining the presence of the disease [15]. HSI was calculated using the following equation: 8 × alanine aminotransferase (ALT, IU/L)/aspartate aminotransferase (AST, IU/L) ratio + BMI (+2, if diabetes; +2, if female). We adopted the cutoff value of HSI 36 to classify NAFLD [15]. In addition, to evaluate the degree of fibrosis among individuals who were diagnosed with NAFLD, the fibrosis-4 (FIB-4) index was utilized [16]. FIB-4 index is a widely used marker to assess liver fibrosis in NAFLD patients without invasive biopsy [16]. This index is calculated using the following formula: age (years) × AST/platelet count (109/L)/ALT^1/2^. The patients were classified into those with FIB-4 value ≤ 1.3 and >1.3 [17]. Although a value of 2.67 would be more predictive in categorizing patients with advanced fibrosis, we dichotomized NAFLD patients using a value of 1.3 because of the small number of data above 2.67. AST and ALT levels were measured using ultraviolet light without the pyridoxal-5-phosphate method (Hitachi Automatic Analyzer Labospect 008AS, Hitachi, Tokyo, Japan).

### 2.3. The STOP-Bang Scoring System

The STOP-Bang model is a convenient triage tool for evaluating OSA probability. The higher the score, the higher the risk of the disease [18]. The sum of the scores ranges from 0 to 8 according to the following eight dichotomous yes/no questionnaires. (1) Do you snore loudly? (2) Do you often feel tired, fatigued, or sleepy during the daytime? (3) Has anyone observed you stop breathing or choking during your sleep? (4) Do you have high blood pressure? (5) Is your BMI > 35 kg/m^2^? (6) Are you over 50 years? (7) Is your neck circumference >40 cm? (8) Are you a male? We then trisected the score into low-, intermediate-, and high-risk of OSA according to the practical guideline for classifying the STOP-Bang score [12]. Individuals with a score of 0–2 were considered to have a low risk of OSA, those who scored 3–4 had an intermediate risk of OSA, and those with a score of 5–8 had a high risk of OSA. However, individuals with a score of 2 along with having other factors, including male sex, BMI > 35, or neck circumference >40 were classified as having a high risk of OSA, although their score was only 3.

### 2.4. Socioeconomic Status, Anthropometric Indices, and Laboratory Results

Data regarding socioeconomic status, such as residence (urban vs. rural), educational level (middle school or lower, high school, vs. college or more), household income level, smoking status (never, former, vs. current), and attainment of adequate physical activity (no vs. yes) were collected. Smoking status was classified based on the questionnaires from the National Health Interview Survey [19]. Adequate physical activity was defined as at least 150 min per week of moderate-intensity aerobic activity or 75 min per week of vigorous aerobic activity. BMI was calculated as weight in kilograms divided by height in meters squared, and a value of 25 kg/m^2^ was used to classify individuals as obese or non-obese.

Other anthropometric indices measured were systolic and diastolic blood pressure, neck circumference, and waist circumference (cm). AST and ALT levels were determined by ultraviolet without pyridoxal-5′-phosphate method using Labospect 008AS (Hitachi/Japan). Fasting glucose (mg/dL) levels were measured by the hexokinase method using Labospect 008AS (Hitachi, Japan). Glycosylated hemoglobin (HbA1c, %) was measured by high-performance liquid chromatography using Tosoh G8 (Tosho/Japan). Cholesterol and triglyceride (mg/dL) levels were determined by an enzymatic method using Labospect 008AS (Hitachi, Japan).

### 2.5. Statistical Analysis

The KNHANES is an annual national survey conducted by the Korea Centers for Disease Control. The KNHANES dataset was constructed using a multistage clustered probability design to represent general noninstitutionalized Korean citizens. Sampling weights were assigned to the sample individuals, and all statistical analyses were performed using complex sample analyses. The detailed study protocol has been previously described [20]. The complete KNHANES dataset is freely available online to the public on the Korea Centers for Disease Control and Prevention website.

Continuous variables were compared using the *t*-test and are presented as the estimated mean value with standard error. Categorical variables were compared using the chi-square or Fisher’s exact test and are presented as an estimated percentage with standard error. The weighted prevalence of NAFLD according to the STOP-Bang score was estimated, and the trend for an increase in prevalence was calculated using the Cochran–Armitage test. Complex sample logistic regression analysis was used to calculate odds ratios (OR) with 95% confidence intervals (CI) for NAFLD and advanced fibrosis. We performed unadjusted, age- and sex-adjusted analysis, and multivariable-adjusted analyses considering residence, education, household income, smoking status, and physical activity as covariables. Subgroup analyses were performed based on sex and obesity status. Additional analyses were performed according to the use of cholesterol-lowering agents and presence of diabetes. All tests were two-tailed, and statistical significance was set at *p* < 0.05. All statistical analyses were performed using the IBM SPSS Statistics for Windows (version 24.0; IBM Corp., Armonk, NY, USA).

## 3. Results

The clinical characteristics of the study participants are summarized in Table 1. The NAFLD group included younger individuals and a higher proportion of men, current smokers, and obese individuals than the group without NAFLD. Regarding metabolic components and laboratory test results, AST, ALT, systolic and diastolic blood pressure, neck and waist circumference, fasting glucose, triglyceride, and high-density lipoprotein cholesterol levels were higher in patients with NAFLD than in those without NAFLD. 

The prevalence of NAFLD increased according to the STOP-Bang score in both men and women (Figure 1). In the groups with OSA risk, the prevalence of NAFLD was 51.4% (men, 51.7%; women, 45.4%). When the participants were classified according to obesity, the trend in which the prevalence of NAFLD increased according to the STOP-Bang score remained significant (Figure 2). In obese individuals at high risk of OSA, the prevalence of NAFLD was 68.3% (68.0% in men and 75.1% in women).

The associations between the risk of OSA and NAFLD in the general population according to sex are summarized in Table 2. Participants with high STOP-Bang scores were more likely to develop NAFLD. In the multivariable-adjusted model, the OR of NAFLD was 5.44 (95% CI: 3.16–9.36) in the high-risk group compared with the low-risk group. The OR for NAFLD was more intense among men. Compared with the low risk of having OSA in men, the OR of NAFLD was 8.49 (95% CI: 3.92–18.37) in the high-risk group.

Subgroup analysis according to obesity status showed a prominent association between the risk of OSA and NAFLD in obese individuals (Table 3). Considering the high risk of OSA in men, the OR of NAFLD was 9.96 (95% CI: 4.32–22.94) in obese individuals and 8.53 (95% CI: 1.79–40.56) in non-obese individuals. However, although a similar trend was also observed in obese women (*p_trend_* < 0.001), the calculated OR of NAFLD did not numerically increase according to the STOP-Bang score.

Appendix A shows an additional subgroup analysis of the relationship between the risk of OSA and NAFLD according to the use of cholesterol-lowering drugs and presence of diabetes. Statistically significant associations were observed in obese individuals who were not taking the drugs and obese participants who did not have diabetes.

Advanced fibrosis, which was defined by a cut-off value of 1.3 on FIB-4, was significantly associated with the STOP-Bang score in the unadjusted model (*p* = 0.004 in obese men and *p* = 0.02 in obese women, Table 4). However, the significance did not remain after adjustment for the multivariable analysis.

## 4. Discussion

This cross-sectional study utilized an easy-to-use, reliable, and effective screening tool, the STOP-Bang questionnaire, to stratify the risk of OSA. We set the STOP-Bang score as an explanatory variable and estimated the prevalence and OR of NAFLD in the general Korean population aged ≥40 years. To the best of our knowledge, this is the first study to evaluate the association between the STOP-Bang stratified risk of OSA and NAFLD in the general population. The incidence of NAFLD was more prominent according to the STOP-Bang score in both sexes. The prevalence of NAFLD was higher among obese than among non-obese individuals, with the rate of approximately 70% in the high-risk STOP-Bang obese group compared with 15% in the non-obese group. We observed a significant association between the STOP-Bang score and the risk of OSA and NAFLD in both sexes. The magnitude of the association was stronger in obese than in non-obese men, whereas the association was only significant in obese women. However, advanced fibrosis was not associated with the STOP-Bang score. These results indicate that OSA is closely related to NAFLD, although further longitudinal prospective studies are needed to demonstrate the relationship between OSA, NAFLD, and advanced fibrosis.

Importantly, obese individuals presented higher STOP-Bang scores, and the association between OSA and NAFLD was more prominent, which is in line with a previous report [21]. In our study, the prevalence of obesity was 68.7%, 42.2%, and 28.8% in the high-risk, intermediate-risk, and low-risk OSA groups, respectively. The association between OSA and NAFLD was more significant in obese individuals. A high BMI is considered an independent risk factor for OSA, and obesity in Asians is more significantly associated with other comorbidities in patients with OSA than in Western people [22]. Yearly weight gain and its velocity in obese patients with OSA were associated with a greater increase in AHI than in non-obese individuals [23]. Meanwhile, intensive lifestyle modifications, including weight reduction, showed an improvement in OSA severity in a 10-year follow-up study [24]. 

Interestingly, as shown in Appendix A, obese individuals not using cholesterol-lowering therapy and without diabetes showed a significant relationship between OSA and NAFLD. Obese individuals not using cholesterol-lowering drugs were younger and had higher income and educational levels than participants using medication (data not shown). Individuals without diabetes were younger, were at a higher rate women, and had higher income, education, and physical activity levels (data not shown). The reason for this observation is unclear, but statins may play a role in the improvement of biopsy-proven NAFLD and may reduce the intensity of the correlation [25].

The STOP-Bang model has been adopted in various sleep-related studies owing to its simplicity and relative convenience. A linear association between the STOP-Bang score and AHI was observed, suggesting the role of the STOP-Bang model in determining OSA severity, detecting unrecognized OSA, and prioritizing patients with high scores for OSA management [18]. However, there is a paucity of data using this model to show its relationship with metabolic disturbances. A cross-sectional study in a military hospital in Saudi Arabia reviewed the records of 306 patients with type 2 diabetes and divided them into low-and high-risk OSA groups according to the STOP-Bang score. The study found a close correlation between a high STOP-Bang score and abdominal obesity, duration of diabetes, and high HbA1c level [26]. Although the STOP-Bang score was utilized in one study of patients with NAFLD to explore the relationship between advanced fibrosis and atherosclerotic disease, only a small number of patients (*n* = 126) were included, and the significance of the association between the STOP-Bang score and the degree of fibrosis did not remain after multivariable adjustment [14]. Therefore, this study is the first to observe a significant relationship between the STOP-Bang score-based stratified risk of OSA and the presence of NAFLD in the general population. 

There is constant interest in the relationship between OSA and NAFLD, although it is multidirectional. IH is one of the most widely studied mechanisms that explain the link between OSA and NAFLD. In an experimental study, IH, which is a hallmark of OSA, was shown to lead to hepatic steatosis by provoking dysregulation of lipid biosynthesis in the liver [27]. The influx of fatty acids into the liver is an important mediator of lipotoxicity, leading to progressive hepatic tissue damage [28]. Nearly two decades ago, a clinical study reported that AST and ALT levels were significantly elevated in patients with OSA [29]. Since then, nearly 20 studies have been published on this relationship in adult human subjects. For example, the nocturnal oxygen desaturation index, which quantifies the severity of chronic IH, was strongly correlated with the degree of NAFLD activity and fibrosis, suggesting a role for sustained IH in OSA patients in the progression of liver injuries [30]. Liver stiffness, measured using transient elastography, showed a dose-dependent relationship with the severity of OSA, with a 7.2-fold increase in significant fibrosis and a 4.7-fold increase in advanced fibrosis [7]. Therefore, it is not surprising that OSA is independently associated with NAFLD. 

IH and other factors may be responsible for the occurrence of NAFLD in patients with OSA. Intrathoracic pressure variation is an OSA-related physiological event, and a sustained pressure swing may result in endothelial dysfunction [31], which leads to various cardiovascular diseases. Sleep fragmentation and recurrent arousals from sleep also induce endothelial dysfunction and recruitment of inflammatory cytokines, such as interleukin-6 [32]. Disrupted endothelial function has been observed in NAFLD patients with advanced fibrosis [33]. 

Although we first evaluated the relationship between the STOP-Bang score-stratified risk of OSA and NAFLD and showed a positive association, especially in the obese population, several limitations should be noted. First, this was a cross-sectional study and temporal causality was not guaranteed; thus, the results should be carefully interpreted. Further longitudinal studies using this simple screening method may provide information about this relationship and help prioritize individuals at high risk of OSA for management. Second, we did not cross-validate the STOP-Bang score with AHI from PSG, which is currently the standard diagnostic method for OSA, because the KNHANES only investigated sleep-related disorders using the STOP-Bang questionnaire from 2019 to 2020. Third, we categorized NAFLD based on noninvasive HSI. However, no clinical or serological examinations have replaced invasive liver biopsy for a definite NAFLD diagnosis. Fourth, because only a small number of women were categorized into NAFLD and STOP-Bang stratified high-risk groups, adequate statistical estimations were unfeasible. Fifth, although we performed subgroup analyses according to sex and BMI, statistical concerns may remain because both the explanatory variable (HSI) and dependent variable (STOP-Bang score) were calculated using sex and BMI. Sixth, considering that a recent study showed that genetic background contributes to the relationship between OSA and triglyceride levels [34], the KNHANES did not assess genetic factors. However, further studies using methods such as Mendelian randomization may provide insights into the observed relationship at the genetic level.

## 5. Conclusions

In conclusion, by analyzing a large sample of Korean adults, this study showed that OSA is closely associated with NAFLD in the general population, although the degree of fibrosis was not associated with OSA severity. In addition, the prevalence of NAFLD and degree of significance of its relationship with the STOP-Bang score were more accentuated in obese population. Our study also suggests that clinicians should pay attention to obese individuals who do not use cholesterol-lowering agents and have diabetes to screen for the co-existence of OSA and NAFLD and to prevent future morbidities.

## Figures and Tables

**Figure 1 ijerph-19-08398-f001:**
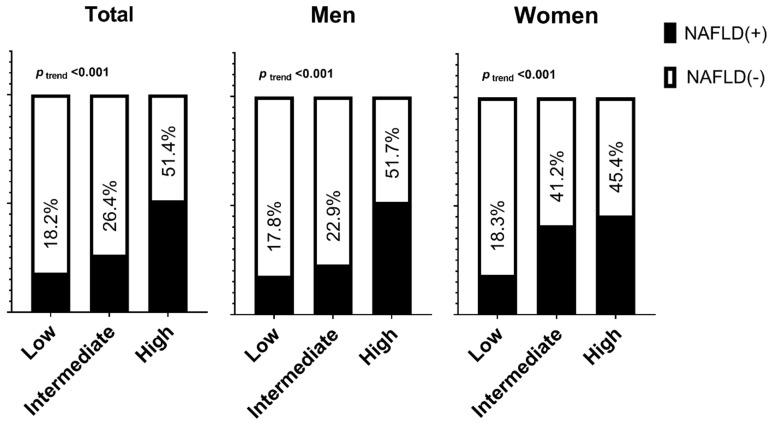
Prevalence of nonalcoholic fatty liver disease according to the risk of having obstructive sleep apnea by sex.

**Figure 2 ijerph-19-08398-f002:**
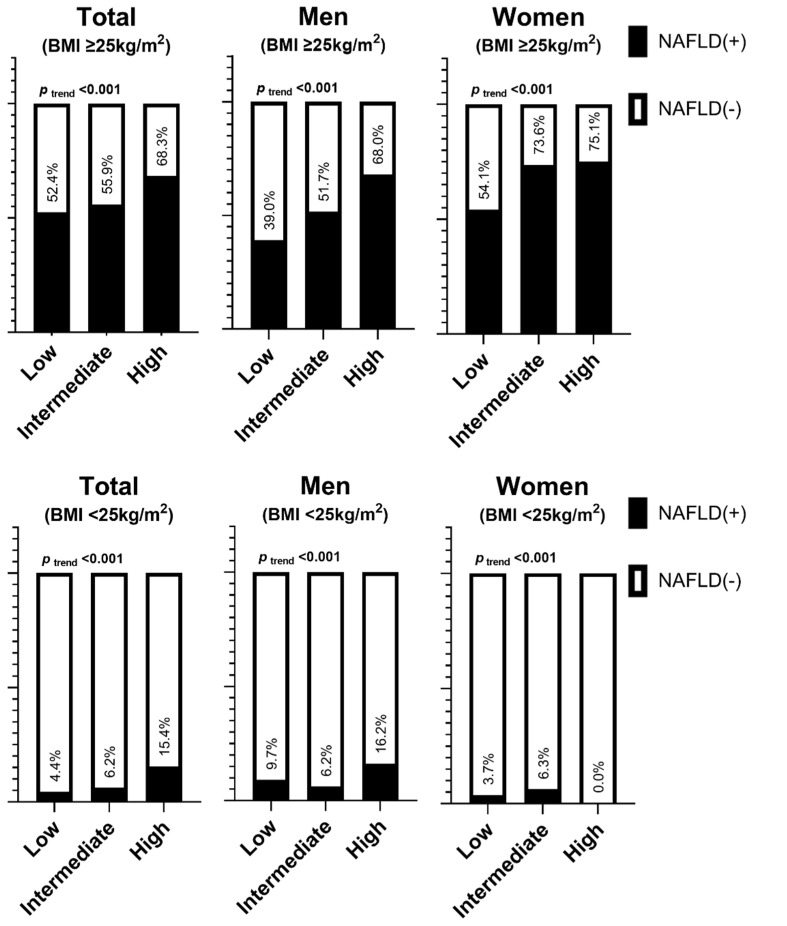
Prevalence of nonalcoholic fatty liver disease according to the risk of having obstructive sleep apnea by sex and obesity status.

**Table 1 ijerph-19-08398-t001:** Characteristics of study participants.

	NAFLD (-) (*n* = 3254)	NAFLD (+) (*n* = 1021)	*p*
**Age (years)**	55.6 (0.3)	53.9 (0.4)	<0.001
**Men**	49.4 (0.9)	56.7 (1.8)	0.001
**Residence**			0.287
Urban	84.9 (1.9)	83.5 (2.2)	
Rural	15.1 (1.9)	16.5 (2.2)	
**Education**			0.454
Middle school or lower	21.0 (1.1)	19.1 (1.4)	
High school	35.2 (1.2)	37.0 (1.9)	
College or more	43.9 (1.6)	43.9 (2.2)	
**Household income**			0.984
Lowest	12.7 (0.8)	12.5 (1.2)	
Middle low	21.6 (1.1)	21.1 (1.6)	
Middle high	29.7 (1.2)	30.2 (1.8)	
Highest	35.9 (1.5)	36.2 (2.0)	
**Smoking**			<0.001
Never	57.2 (1.0)	50.1 (1.9)	
Former	26.4 (1.0)	27.2 (1.7)	
Current	16.4 (0.8)	22.7 (1.7)	
**Adequate Physical activity ¶**			0.073
No	56.6 (1.0)	60.5 (1.9)	
Yes	43.4 (1.0)	39.5 (1.9)	
**BMI (kg/m^2^)**			<0.001
<18.5	3.0 (0.4)	-	
18.5–22.9	44.1 (1.1)	2.4 (0.6)	
23–24.9	30.7 (0.9)	11.3 (1.2)	
≥25	22.2 (0.9)	86.3 (1.2)	
**AST (IU/L)**	23.4 (0.2)	27.3 (0.4)	<0.001
**ALT (IU/L)**	19.1 (0.2)	36.5 (0.8)	<0.001
**SBP (mmHg)**	118.9 (0.4)	123.9 (0.5)	<0.001
**DBP (mmHg)**	76.4 (0.2)	80.7 (0.4)	<0.001
**Neck circumference (cm)**	34.9 (0.1)	37.8 (0.1)	<0.001
**Waist circumference (cm)**	82.3 (0.2)	94.4 (0.3)	<0.001
**Fasting glucose (mg/dL)**	99.6 (0.4)	113.4 (1.1)	<0.001
**HbA1c (%)**	5.74 (0.02)	6.26 (0.04)	<0.001
**HDL cholesterol (mg/dL)**	53.4 (0.3)	46.5 (0.3)	<0.001
**LDL cholesterol (mg/dL)**	116.2 (2.2)	119.1 (2.5)	0.399
**Triglyceride (mg/dL)**	127.7 (2.1)	178.3 (4.8)	<0.001

Data are presented as estimated mean (SE) for continuous variables and estimated percentage (SE) for categorical variables, otherwise stated. **¶** Adequate physical activity was defined as at least 150 min per week of moderate-intensity aerobic activity or 75 min per week of vigorous aerobic activity. NAFLD = non-alcoholic fatty liver disease, BMI = body mass index, AST = aspartate aminotransferase, ALT = alanine aminotransferase, SBP = systolic blood pressure, DBP = diastolic blood pressure, HbA1c = glycated hemoglobin, HDL = high density lipoprotein, SE = standard error.

**Table 2 ijerph-19-08398-t002:** Relationship between the risk of having OSA and NAFLD by sex.

	Crude		Age and (Sex) Adjusted		Multivariable-Adjusted	
	OR (95% CI)	*p*	OR (95% CI)	*p*	OR (95% CI)	*p*
**All**		<0.001		<0.001		<0.001
Low	1		1		1	
Intermediate	1.76 (1.48–2.09)		3.93 (2.97–5.21)		2.59 (1.89–3.56)	
High	5.12 (3.64–7.21)		12.79 (8.15–20.09)		5.44 (3.16–9.36)	
**Men**		<0.001		<0.001		<0.001
Low	1		1		1	
Intermediate	1.63 (1.03–2.58)		3.40 (1.78–6.49)		3.44 (1.79–6.61)	
High	5.48 (3.18–9.46)		8.63 (4.01–18.57)		8.49 (3.92–18.37)	
**Women**		<0.001		<0.001		<0.001
Low	1		1		1	
Intermediate	3.25 (2.46–4.29)		2.19 (1.53–3.16)		2.13 (1.49–3.05)	
High	4.57 (1.10–18.99)		2.69 (0.42–17.32)		2.37 (0.36–15.48)	

Multivariate model considered age, (sex), residence, education, household income, smoking, physical activity, and body mass index as covariates. The risk of having OSA was measured using STOP-Bang questionaries. OSA = obstructive sleep apnea, NAFLD = non-alcoholic fatty liver disease, STOP-Bang = snoring, tiredness, observed apnea, high blood pressure, body mass index, age, neck circumference, and male gender, OR = odds ratio, CI = confidence interval.

**Table 3 ijerph-19-08398-t003:** Relationship between the risk of having OSA and NAFLD by sex and obese status.

	Crude		Age-Adjusted		Multivariable-Adjusted	
	OR (95% CI)	*p*	OR (95% CI)	*p*	OR (95% CI)	*p*
**Men**						
**BMI ≥ 25 kg/m^2^**		<0.001		<0.001		<0.001
Low	1		1		1	
Intermediate	1.67 (0.88–3.19)		4.10 (2.06–8.16)		4.29 (2.12–8.71)	
High	3.33 (0.58–7.04)		9.76 (4.33–22.04)		9.96 (4.32–22.94)	
**BMI < 25 kg/m^2^**		0.687		0.003		0.008
Low	1		1		1	
Intermediate	0.66 (0.29–1.50)		2.96 (0.86–10.23)		2.76 (0.82–9.36)	
High	1.93 (0.59–6.34)		10.44 (2.23–48.95)		8.53 (1.79–40.56)	
**Women**						
**BMI ≥ 25 kg/m^2^**		<0.001		<0.001		<0.001
Low	1		1		1	
Intermediate	2.36 (1.56–3.58)		2.83 (1.81–4.43)		2.71 (1.74–4.23)	
High	2.56 (0.28–23.39)		3.23 (0.44–23.74)		2.60 (0.34–19.70)	
**BMI < 25 kg/m^2^**		0.161		0.853		0.837
Low	1		1		1	
Intermediate	1.74 (0.86–3.51)		1.13 (1.13–1.13)		0.927 (NA–NA)	
High	NA (NA–NA)		NA (NA–NA)		NA (NA–NA)	

Multivariate model considering age, residence, education, household income, smoking, physical activity, and body mass index as covariates. The risk of having OSA was measured using STOP-Bang questionaries. OSA = obstructive sleep apnea, NAFLD = non-alcoholic fatty liver disease, STOP-Bang = snoring, tiredness, observed apnea, high blood pressure, body mass index, age, neck circumference, and male gender, OR = odds ratio, CI = confidence interval.

**Table 4 ijerph-19-08398-t004:** Relationship between the risk of having OSA and advanced fibrosis assessed by FIB-4 in NAFLD patients.

	FIB-4 ≤ 1.3 vs. >1.3					
	Crude		Age-Adjusted		Multivariable-Adjusted	
	OR (95% CI)	*p*	OR (95% CI)	*p*	OR (95% CI)	*p*
**Men**						
**All**		0.005		0.157		0.135
Low	1		1		1	
Intermediate	2.42 (0.54–10.79)		0.42 (0.05–3.91)		0.39 (0.04–3.48)	
High	4.71 (0.99–22.47)		0.78 (0.08–7.57)		0.75 (0.08–6.90)	
**BMI ≥ 25 kg/m^2^**		0.004		0.191		0.189
Low	1		1		1	
Intermediate	2.68 (0.34–20.84)		0.36 (0.04–3.31)		0.33 (0.04–3.16)	
High	5.46 (0.68–44.08)		0.68 (0.07–6.78)		0.65 (0.07–6.43)	
**BMI < 25 kg/m^2^**		0.602		0.961		0.191
Low	1		1		1	
Intermediate	1.99 (0.23–17.14)		1.01 (0.46–2.18)		10.34 (0.04–2698.53)	
High	1.97 (0.11–35.43)		0.96 (0.12–7.86)		1.48 (0.01–298.27)	
**Women**						
**All**		0.015		0.458		0.831
Low	1		1		1	
Intermediate	2.07 (1.30–3.28)		0.89 (0.51–1.58)		1.01 (0.56–1.81)	
High	0.43 (0.04–4.23)		0.28 (0.02–4.01)		0.46 (0.04–4.86)	
**BMI ≥ 25 kg/m^2^**		0.02		0.487		0.837
Low	1		1		1	
Intermediate	2.08 (1.29–3.33)		0.91 (0.49–1.68)		1.02 (0.54–1.93)	
High	0.43 (0.04–4.23)		0.26 (0.02–3.96)		0.44 (0.04–4.85)	
**BMI < 25 kg/m^2^**		0.468		0.489		0.844
Low	1		1		1	
Intermediate	1.78 (0.36–8.77)		0.58 (0.12–2.80)		0.80 (0.08–7.78)	
High	NA (NA–NA)		NA (NA–NA)		NA (NA–NA)	

Multivariate model considering age, residence, education, household income, smoking, physical activity, and body mass index as covariates. The risk of having OSA was measured using STOP-Bang questionaries. OSA = obstructive sleep apnea, NAFLD = non-alcoholic fatty liver disease, FIB-4 = Fibrosis-4, STOP-Bang = snoring, tiredness, observed apnea, high blood pressure, body mass index, age, neck circumference, and male gender, OR = odds ratio, CI = confidence interval.

## Data Availability

The data of the KNHANES is opened to the public, therefore, any researcher can be obtained after request from the website https://knhanes.kdca.go.kr/knhanes/main.do (accessed on 1 May 2022).

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
