# Peer review of "Obstructive Sleep Apnea and Nonalcoholic Fatty Liver Disease in the General Population: A Cross-Sectional Study Using Nationally Representative Data"

_ijerph, 2022, doi:10.3390/ijerph19148398_

Round 1
Reviewer 1 Report
The authors evaluated the association between obstructive sleep apnea (OSA) and nonalcoholic fatty liver disease (NAFLD) in the general population using a nationally representative 16 sample from South Korea. This is a well-conducted study. The only concern is that the authors did not describe the criteria of NAFLD. NAFLD is a progressive disease and criteria need to be defined. Inflammation is one of the criteria. Yet the authors did not assess inflammatory markers.
Reviewer 2 Report
In addition to confirming the existing correlation between OSA and NAFL, previously analyzed in [7] and [21], the main contribution of this work is to stratify the risk levels of suffering from OSA and NAFLD, and rule out a possible relationship with advanced fibrosis.
The statistical analysis performed on the data seems adequate to me, the article is well structured, I would only consider adjusting the size of the graphs in Figures 1 and 2, they look very large.
Reviewer 3 Report
I have read the article by Kim et al with great interest. The authors investigated the relationship between OSA risk and NAFLD in a large Korean population.
Comments:
· 1. Abstract. Line 19. Please, change “male sex” to “gender”.
· 2. Abstract. Conclusion. Rather than repeating the results, I suggest adding something about the clinical relevance of the study (i.e. patients with high STOP-BANG score should be screened for NAFLD).
· 3. Methods. 2.2. Please, add how HSI was calculated.
· 4. Results. Do you have any data on lipid lowering medications? Patients with and without these medications should be analysed separately.
· 5. Results. Please, provide data on the prevalence of diabetes. Patients with and without diabetes should be analysed separately.
· 6. Discussion. Both STOP-BANG and HSI incorporate gender and BMI. Therefore, the relationship between them is not surprising, but evident. Please, discuss.
· 7. Discussion. The association between OSA and triglyceride levels (i.e. de novo hepatic lipid synthesis) is determined by genetic factors (https://pubmed.ncbi.nlm.nih.gov/31908118/). Taking into account the fact that this study was performed in a Korean population, the data may not necessarily be generalised. Please, discuss.
Round 2
Reviewer 1 Report
I have no other comment
Reviewer 3 Report
I am happy with the changes and suggest acceptance.